# Improving CNN training by Riemannian optimization on the generalized Stiefel manifold combined with a gradient-based manifold search

## Abstract

Enforcing orthonormality constraints in deep learning has been shown to provide significant benefits. Although hard restrictions can be applied by constraining parameter matrices to the Stiefel manifold, this approach limits the solution space to that specific manifold. We show that a generalized Stiefel constraint $X^T SX = \mathbb{I}$ for Riemannian optimization can lead to even faster convergence than in previous work on CNNs, which enforced orthonormality. The gained flexibility comes from a larger search space. In this paper, we therefore propose a novel approach that retains the advantages of compact restrictions while using a gradient-based formulation to adapt the solution space defined by $S$. This approach results in overall faster convergence rates and improved test performance across CIFAR10, CIFAR100, SVHN, and Tiny ImageNet32 datasets on GPU hardware.

## 1 Introduction

The incorporation of orthonormality constraints in deep learning has gained considerable interest due to its potential benefits during training. For example, Bansal et al. (2018) have shown that using orthogonality regularizations within the training of convolutional neural networks (CNNs) can improve both the final accuracy and lead to faster and more stable convergence. Furthermore, Huang et al. (2018) demonstrated that rectangular orthogonal matrices can stabilize the distribution of network activations in feed-forward neural networks and Arjovsky et al. (2016) showed that by constructing unitary weight matrices in recurrent neural networks (RNNs), they can mitigate vanishing and exploding gradients.

One method to ensure orthonormality for parameter matrices is to use Riemannian optimization techniques on the Stiefel manifold. The Stiefel manifold is the set of all orthonormal $p$-frames in an $n$-dimensional Euclidean space. As Riemannian optimization is generally time-consuming, Li et al. (2020) approximated the closed-form by an iterative version. Using this iterative version, they have shown theoretically and empirically that the training of CNNs can be accelerated on modern GPUs.

Although the orthonormal Stiefel manifold restriction already yields many benefits for deep learning, it also reduces the solution space of the $n \times p$ parameter matrices to a subspace of the Euclidean subspace of size $np - \frac{p(p+1)}{2}$. Vorontsov et al. (2017) have shown that while orthogonal initialization is beneficial in their experiments, enforcing strict orthogonality constraints can also be disadvantageous. Instead of using orthormality only in the initialization phase, we propose an approach to generalize the Stiefel manifold during training in order to better fit the training data set. Instead of using only strict orthonormal matrices, the generalized form of the Stiefel manifold, which we use in this paper, is defined by an overlap matrix $S$. The union over all overlap matrices is the space of all full-rank matrices. This means that a flexible overlap matrix has the potential to greatly expand the solution space. Although such generalized Stiefel manifolds have been of increasing interest in recent years Sato & Aihara (2019); Li et al. (2021); Sedano-Mendoza (2022); Shustin & Avron (2023), to the best of our knowledge, it has, in contrast to the non-generalized Stiefel manifold efficiently used by Li et al. (2020), not yet been implemented on CNNs.

We introduce a novel approach that restricts the parameter matrices to the set of generalized Stiefel manifolds and dynamically optimizes the overlap matrix $S$ (which completely determines this manifold) using gradient-based optimization. Our optimization procedure builds on two novel parts, an adapted Riemannian optimization procedure on the generalized Stiefel manifold given an overlap matrix and a gradient-based optimization procedure of the generalized Stiefel manifold itself. We use our generalized version of the Riemannian optimization to insure the inherited beneficial properties of the orthonormal approach, since we still optimize on a compact set. The generalization leads to a much larger search space. Moving this hyper-parameter optimization problem into a gradient-based optimization, we manage to tackle this problem efficiently without artificially restricting $S$. Our results demonstrate improved empirical convergence rates and overall accuracy on not only for specific manifolds but also that finding a fitting manifold can be efficiently realized within the same CNN training framework using standard optimizers like Adam.

The paper is structured as follows: First, we will present related work on Riemannian optimization as well as work on the generalized Stiefel manifold. Then, we will formally introduce the generalized Stiefel manifold and some of its properties, as well as the generalization of the Cayley transformation. Next, we discuss our optimization approach for $S$ that leads to our final strategy to train the CNNs. In a series of empirical experiments, we can show that the additional complexity of hyperparameter optimization is outweighed by the faster convergence rates and higher test accuracies. Finally, we discuss limitations and future research of this aproach.

## 2 RELATED WORK

A major challenge in Riemannian optimization is translation from one point on the manifold to another. A standard solution is exponential mapping. However, for the Stiefel manifold, this is computationally infeasible in most cases. There are many different retraction mappings Absil et al. (2008). One of the more popular choices is the Cayley transformation, which in its closed form requires a computationally expensive matrix inversion. However, a recent method uses an iterative approximation Li et al. (2020), which is much more efficient and also the choice for our optimization procedure. There are also other approaches to reduce computational cost, but they are restrictive, such as e.g. Wen & Yin Wen & Yin (2013), whose approach is efficient only when the dimensions of the Stiefel matrix are far apart.

Optimization on the generalized Stiefel manifold and its properties has received increasing interest in recent years. Cholesky et al. Sato & Aihara (2019) generalized the QR-retraction to the generalized Stiefel manifold and showed that it is more effective than retractions based on polar factorization. Shustin & Avron Shustin & Avron (2023) have used Riemannian preconditioning to overcome shortcomings of standard geometric components, such as slow convergence. Moreover, Sedano-Mendoza Sedano-Mendoza (2022) has analyzed isometry groups of the generalized Stiefel manifold by considering a non-associative algebra associated to the structure of the Stiefel manifold. However, to the best of our knowledge there is no research of the impacts of implementing an optimization procedure on the generalized Stiefel manifold for deep learning.

## 3 PRELIMINARY

This section gives an overview of the generalized Stiefel manifold, the generalization of the Cayley transformation, and discusses the construction of the overlap matrix $S$. A deeper inside is given by Shustin & Avron (2023).

### 3.1 THE GENERALIZED STIEFEL MANIFOLD

**Definition 1** *(Riemannian manifold) A Riemannian manifold $(\mathcal{M}, g)$ is a smooth manifold $\mathcal{M}$ equipped with a Riemannian metric $g$. The Riemannian metric $g$ is a smooth assignment of an inner product $g_p$ on the tangent space $T_p\mathcal{M}$ at each point $p \in \mathcal{M}$.*

**Definition 2** *(Retraction) A retraction on a manifold $\mathcal{M}$ is a smooth map $R_x : T_x\mathcal{M} \to \mathcal{M}$ from the tangent space $T_x\mathcal{M}$ at the point $x \in \mathcal{M}$ to the manifold $\mathcal{M}$ itself. For each point $x \in \mathcal{M}$, the retraction map $R_x : T_x\mathcal{M} \to \mathcal{M}$ satisfies the following conditions: $R_p(0_p) = p$ and $dR_p(0_p)$ is the identity map on $T_p\mathcal{M}$, where $0_p$ is the zero vector in $T_p\mathcal{M}$.*

**Definition 3** *(Generalized Stiefel Manifold) The generalized Stiefel manifold is given by*

$$St_S(n, p) = \{X \in \mathbb{R}^{n \times p} : X^T S X = \mathbb{I}\}$$

*where $n, p \in \mathbb{N}$ with $n \geq p$ and $S \in \mathbb{R}^{n \times n}$ being a symmetric matrix. The Stiefel manifold $St(n, p)$ is the generalized Stiefel manifold iff $S = \mathbb{I}$.*

The generalized Stiefel manifold is given by definition 3. For our purposes, we also restrict the overlap matrix $S$ to be strictly positive definite, which in particular means that it has full rank. Furthermore, the generalized Stiefel manifold is a Riemannian manifold and an embedded submanifold of a Euclidean space. One can define its Riemannian metric by $g(Z_1, Z_2) = \mathrm{tr}(Z_1^T S Z_2)$. Moreover, its tangent space is given by

$$T_x St_S(n, p) = \{Z \in \mathbb{R}^{n \times p} : Z^T S X + X^T S Z = 0\}$$

Another important mapping for optimization is the projection mapping. It can be calculated as follows:

$$\pi_{T_x}(Z) = WSX, \text{where } W = \hat{W} - \hat{W}^T \tag{1}$$

with

$$\hat{W} = ZX^T - \frac{1}{2} X \left( X^T S Z X^T \right)$$

Our optimization procedure of the objective function $f(X)$ follows the optimization procedure of Li et al. Li et al. (2020). It can be separated into the following steps. First, we calculate the gradient in the Euclidean space $\nabla f(X)$ and linearly combine it with any additional terms such as, e.g., the momentum. Then we project the linear combination onto the tangent space $T_X St(n, p)$ using equation 1. Note that the order can also be reversed: First, project the gradient and the additional terms and then combine them linearly, since the projection mapping is linear. The projected gradient can be written as $\nabla_{St} f(X)$. Finally, using the previously calculated tangent vector, we calculate the new point on the manifold. Since other methods like the exponential map or parallel transport are computationally infeasible, we use the Cayley transformation, which will be discussed in the next subsection.

### 3.2 THE ADAPTED CAYLEY TRANSFORMATION

The Cayley transformation is a retraction map on the Stiefel manifold. We need to adapt it slightly to generalize it to the generalized Stiefel manifold. It is given by:

$$Y(\alpha) = \left(\mathbb{I} - \frac{\alpha}{2} WS\right)^{-1} \left(\mathbb{I} + \frac{\alpha}{2} WS\right) X \tag{2}$$

where $X \in St_S(n, p)$, $W \in \mathbb{R}^{n \times n}$ is skew-symmetric ($W^T = -W$) and $\alpha$ is a parameter that represents the step size. One can easily verify that this is indeed a retraction given by definition 2 by showing $Y(0) = X$ and $\frac{d}{d\alpha} Y(0) = WSX$. Even though we can assume that $Y(\alpha) \in St_S(n, p)$ if $X \in St_S(n, p)$, there is a proof of this in Appendix A. By choosing $W$ corresponding to the projection mapping in equation 1, the Cayley transformation implicitly initially projects the gradient on the tangent space. The Cayley transformation has the advantage that, by using its iterative form, it is computationally inexpensive. The closed form still has an expensive matrix inversion part. The iterative form is given by a slight modification of the fixed-point form, which can be obtained by rearranging equation 2. The fixed-point form and the iterative form are given by

$$Y(\alpha) = X + \frac{\alpha}{2} WS(x + Y(\alpha)), \quad Y^{i+1} = X + \frac{\alpha}{2} WS\left(X + Y^i\right). \tag{3}$$

Li et al. Li et al. (2020) show that for $S = \mathbb{I}$ the iterative Cayley transformation converges to the closed form. Formally, for $\alpha \in (0, \min\{1, 2/\|W\|\})$, the iterative Cayley transformation given by equation 3 is a contraction mapping and converges to the closed form given by equation 2 under the condition of Lipschitz continuity of the gradient of the objective function. This can be generalized for a general $S$ as a symmetric, positiv definite matrix analogously to their proof. We only need to adapt $\alpha \in (0, \min\{1, 2/\|WS\|\})$.

## 4 ALGORITHMS

**Algorithm 1** Cayley SGD with momentum on the generalized Stiefel manifold

**Input:** learning rate $l$, momentum coefficient $\beta$, $\epsilon = 10^{-8}$, $q = 0.5$, $s = 5$
Initialize $S = R^\top R$ and $\hat{X}$ as an orthonormal matrix; $M_1 = 0$
$X_1 = R^{-1}\hat{X}$
**for** $k = 0$ **to** $T$ **do**

$\qquad M_{k+1} \leftarrow \beta M_k - \nabla X_k$

$\qquad \hat{W}_k \leftarrow M_{k+1}X_k - \frac{1}{2}X_k(X_k^\top S M_{k+1}X_k)$
$\qquad W_k \leftarrow \hat{W}_k - \hat{W}_k^\top$
$\qquad M_{k+1} \leftarrow W_k S X_k$
$\qquad \alpha \leftarrow \min\left\{l, \frac{2q}{\|W_k S\| + \varepsilon}\right\}$
$\qquad$ Initialize $Y^0 \leftarrow X_k + \alpha M_{k+1}$
$\qquad$ **for** $i = 1$ **to** $s$ **do**
$\qquad\qquad Y^i \leftarrow X_k + \frac{\alpha}{2}W_k S(X_k + Y^{i-1})$
$\qquad$ **end for**
$\qquad$ Update $X_{k+1} \leftarrow Y^s$
**end for**

**Algorithm 2** Cayley ADAM on the generalized Stiefel manifold

**Input:** learning rate $l$, momentum coefficients $\beta_1$ and $\beta_2$, $\epsilon = 10^{-8}$, $q = 0.5$, $s = 5$
Initialize $S = R^\top R$ and $\hat{X}$ as an orthonormal matrix; $M_1 = 0$, $v_1 = 1$
$X_1 = R^{-1}\hat{X}$
**for** $k = 0$ **to** $T$ **do**
$\qquad M_{k+1} \leftarrow \beta_1 M_k + (1 - \beta_1)\nabla X_k$
$\qquad v_{k+1} \leftarrow \beta_2 v_k + (1 - \beta_2)\|\nabla X_k\|^2$
$\qquad \hat{v}_{k+1} \leftarrow \frac{v_{k+1}}{(1 - \beta_2^k)}$
$\qquad r \leftarrow \frac{(1 - \beta_1^k)}{\hat{v}_{k+1}}$
$\qquad \hat{W}_k \leftarrow M_{k+1}X_k - \frac{1}{2}X_k(X_k^\top S M_{k+1}X_k)$
$\qquad W_k \leftarrow \frac{(\hat{W}_k - \hat{W}_k^\top)}{r}$
$\qquad M_{k+1} \leftarrow W_k S X_k$
$\qquad \alpha \leftarrow \min\left\{l, \frac{2q}{\|W_k S\| + \varepsilon}\right\}$
$\qquad$ Initialize $Y^0 \leftarrow X_k - \alpha M_{k+1}$
$\qquad$ **for** $i = 1$ **to** $s$ **do**
$\qquad\qquad Y^i \leftarrow X_k - \frac{\alpha}{2}W_k S(X_k + Y^{i-1})$
$\qquad$ **end for**
$\qquad$ Update $X_{k+1} \leftarrow Y^s$
**end for**

This section introduces the algorithms used to restrict CNN training to the set of generalized Stiefel manifolds. First, generalized versions of the Cayley SGD with momentum and the Cayley ADAM are presented. Second, the gradient-based search for $S$ is discussed. Both are contributions of this paper.

### 4.1 OPTIMIZATION OF THE GENERALIZED STIEFEL MAINFOLD

The following section describes the optimization procedures. We have modified two different algorithms from Li et al. Li et al. (2020), namely the Cayley SGD with momentum and the Cayley ADAM. Both algorithms require that the initial state is an element of the generalized Stiefel manifold. Similar to Li et al., we use the QR decomposition Absil et al. (2008) to initialize the first state. The QR decomposition returns a matrix on the regular Stiefel manifold. Using the property $S = R^T R$, we can set the initial state to

$$X_1 = R^{-1}Q$$

where Q is the Stiefel part of the QR-decomposition. Therefore, $X_1$ is an element of $\text{St}_S(n, p)$

### 4.1.1 CAYLEY SGD WITH MOMENTUM

Stochastic gradient descent Robbins & Monro (1951) with momentum is a widely used optimization method in machine learning. We adapt the algorithm so that it optimizes only on the generalized Stiefel manifold. The updated momentum is computed by projecting the linear combination of the momentum of the previous step and the Euclidean gradient:

$$M_{k+1} = \pi_{T_{X_k}}(\beta M_k - \nabla X_k)$$

here $\beta$ is a hyperparameter. Note that this is equivalent to projecting the momentum and the gradient before linearly combining since the projecting map itself is linear. The new momentum $M_{k+1}$ gives the direction. By applying the Cayley transformation given by equation 3, we calculate the next point on the manifold. The detailed algorithm is presented as algorithm 1.

### 4.1.2 CAYLEY ADAM

ADAM Kingma & Ba (2014) is a commonly used optimization algorithm. The first-order gradient-based algorithm is based on adaptive estimates of lower-order moments. Like the SGD, we adapt the algorithm to restrict $X$ to the generalized Stiefel manifold. The procedure is similar to the modification of the SGD and its details can be reviewed in algorithm 2.

### 4.2 GRADIENT-BASED MANIFOLD OPTIMIZATION

We use a gradiend-based optimization of the generalized Stiefel manifold. Since the overlap matrix $S$ is symmetric and positive definite, it can be decomposed into $S = R^T R$ where $R \in \mathbb{R}^{n \times n}$. With this property we can define a map $X_i \in \text{St}_{S_i}(n, p)$ to $X_{i+1} \in \text{St}_{S_{i+1}}(n, p)$ with the following equation:

$$X_{i+1} = R_{i+1}^{-1} R_i X_i \qquad (4)$$

This is a straightforward way to update the kernel matrices $X$ to a new Stiefel restriction. We apply this update of the weight matrices at the beginning of the forward pass for given $R_i$ and $X_i$. With this relationship between weight matrices under different generalized Stiefel restrictions, we can compute the gradient of the loss with respect to the new matrix $R_{i+1}$ during backpropagation. $R_{i+1}$ then defines the new generalized Stiefel matrix. During this procedure the weights are only updated using equation 4.

Note that this procedure expands our solution space to every full-rank matrix. A larger solution space can be advantageous, but the set of all full-rank matrices is no longer compact. One solution is to mainly use the optimization process with a constant $S$. Another method could be to regularize of the loss function with a term of the form $\|S - \mathbb{I}\|^\mu$ with a large $\mu$. However, this loss regularization was not necessary for our experiments due to a small learning rate and the predominant use of the Riemannian optimization for a constant $S$.

The disadvantage of this procedure is the computationally expensive calculations of the inverse matrix. One epoch takes a little more than twice as long as the Riemannian optimization with a constant $S$.

## 5 EXPERIMENTS

In this section, we present the experiments of our method. First, we present the datasets, models, and parameter choices for the experiments. Then, we discuss our training strategy. Finally, we discuss the impact of our method on training convergence, test performance, and the behaviour over time.

### 5.1 DATASETS, MODELS AND PARAMETER CHOICES

CNN filter can be represented by the matrix $K \in \mathbb{R}^{p \times n}$, where $p = c_\text{out}$ is the output dimension of the filter and $n = c_\text{in} \cdot h \cdot w$, where $c_\text{in}$ is the input dimension and $h$ and $w$ are respectively the height and width of the filter. If $n \geq p$, which is the case most of the time, then the convolutional layer is constrained to $\text{St}_S(n, p)$. In all other cases, we use the standard SGD or standard ADAM.

To obtain comparable results, we chose our models and our parameters similar to the choices of Li et al. Li et al. (2020). We used four datasets, the CIFAR 10 and CIFAR 100 datasets Krizhevsky et al. (2009), the SVHN dataset Netzer et al. (2011) and the Tiny ImageNet dataset Le & Yang (2015) with size-reduced images to 32×32. The CIFAR datasets contains 60,000 32 × 32 color images which are equally divided into 10 and 100 classes, respectively. There are 50,000 training images and 10,000 test images. The SVHN data set contains 73,257 training images and 26032 test images divided into 10 classes. Lastly, Tiny ImageNet32 contains 100,000 training images and 10,000 test images equally divided into 200 classes. For computational feasibility, we reduced the original size of 64×64 color images to 32×32 using box filter, which is used to reduce the size of the ImageNet Deng et al. (2009) itself by the authors.

Furthermore, we used a Wide ResNet Zagoruyko & Komodakis (2016) of depth 28 and width 10 and a VGG Simonyan & Zisserman (2014) with depth 16. The basic structure of both models consists of

convolutional layers of size $3 \times 3$ followed by batch normalization and ReLU activation functions. To train the network during the optimization with the generalized Cayley transformation, we used two different learning rates of 0.01 and 0.1 for the Euclidean space layers and the generalized Stiefel manifold layers, respectively. During the manifold search we used a the standard ADAM optimizer with a learning of 0.00005. Furthermore, we used a batch size of 128, a weight decay of 0.0005 and a learning rate decay of the factor 0.2 at epochs 60, 120 and 160. The maximum number of epochs was 200. The overlap matrix was initialized as the unit matrix, i.e. $S = \mathbb{I}$. As a baseline, we did our experiments without a manifold search to reproduce the results with the regular Stiefel manifold.

## 5.2 TRAINING STRATEGY

Our training consists of two parts, the Riemannian optimization procedure on the generalized Stiefel manifold for a fixed overlap matrix and the gradient-based optimization of the overlap matrix itself. This leaves flexibility in the order in which each part is applied during the training procedure. We have chosen the following. We start by optimizing the overlap matrix for twenty epochs and then switch to Riemannian optimization with a fixed overlap matrix $S$ for the remaining epochs. The idea for this approach has several reasons. One reason is that we first try to find a good solution space determined by the overlap matrix and then optimize with the resulting manifold until the end. Furthermore, we want to minimize the switching between the two algorithms, since the search spaces of the two algorithms are fundamentally different. Each switch changes the landscape of local minima, which works against finding them quickly. In addition, we want to finish the procedure with one algorithm for a majority of epochs to run into the local minimum. Therefore, we minimized the number of switches to one. Another aspect is time efficiency, which will be discussed in more detail in the 5.4 section. Since the manifold search takes more time, it is preferable to apply it to a minority of epochs.

## 5.3 CONVERGENCE ANALYSIS

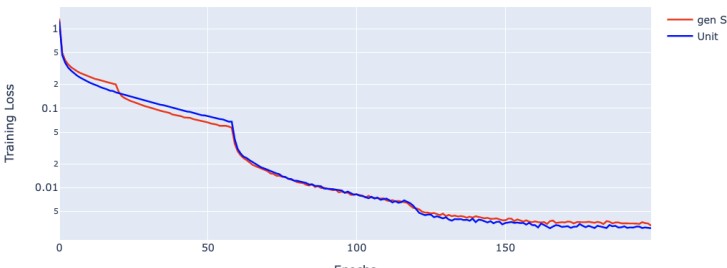

Figure 1: Logarithmically scaled training loss displayed for the entire training process with a Wide ResNet on the SVHN dataset with the generalized Cayley SGD approach. The blue lines represent the baseline approach and the red line our approach with the manifold optimization and the generalized Stiefel manifold restriction.

The training loss for all 200 epochs is shown in figure 1. Here, the procedure was applied to the SVHN dataset using the generalized Cayley SGD method. The red line represents the baseline approach with $S = \mathbb{I}$ for the entire training. The blue line is our approach. For the first twenty epochs, the baseline approach shows better convergence because the manifold optimization does not optimize the loss as efficiently as the Riemannian method. However, when we switch to the Riemannian method at epoch 20, the loss converges faster than the baseline approach. This shows that a solution space with better convergence properties has been found. The training loss shows similar behavior for the other datasets for both generalized Cayley optimization methods.

## 5.4 PERFORMANCE

Table 1 shows the classification error on the test set for all four datasets. Eight different experiments with two different approaches are shown for each dataset. These were run three times each to

Table 1: Classification test error in % of all datasets

| MODEL | OPTIMIZER | $S$ | CIFAR10 | CIFAR100 | SVHN | TINY IMAGENET |
|-------|-----------|-----|---------|----------|------|---------------|
| WRN | CAYLEY SGD | UNIT | $3.73 \pm 0.17$ | $18.35\pm0.22$ | $3.41\pm0.06$ | $47.09\pm 0.33$ |
| | | GEN ST | $3.34 \pm 0.06$ | $17.33\pm0.10$ | $2.97\pm0.01$ | $44.99\pm 0.52$ |
| | CAYLEY ADAM | UNIT | $3.62\pm0.07$ | $18.3\pm0.16$ | $2.84\pm0.13$ | $45.79\pm0.17$ |
| | | GEN ST | $3.61\pm0.11$ | $18.1\pm0.21$ | $2.63\pm0.11$ | $44.66\pm0.15$ |
| VGG | CAYLEY SGD | UNIT | $5.81 \pm 0.19$ | $25.70\pm0.25$ | $3.78\pm 0.10$ | $60.98\pm 0.35$ |
| | | GEN ST | $5.78\pm0.24$ | $23.95\pm0.38$ | $3.74\pm0.10$ | $57.34\pm0.29$ |
| | CAYLEY ADAM | UNIT | $5.97\pm 0.17$ | $25.53\pm 0.33$ | $3.44\pm0.11$ | $57.93\pm0.11$ |
| | | GEN ST | $5.95\pm 0.15$ | $25.45\pm0.23$ | $3.41\pm 0.04$ | $56.72\pm0.81$ |

calculate the standard deviation. The first four experiments show the results for the Wide ResNet and the last four for the VGG. The four experiments are further divided by the optimization method (generalized Cayley SGD and generalized Cayley ADAM). The baseline approach is denoted by Unit and our approach is denoted by gen St. The Unit approach shows the results for $S = \mathbb{I}$, which makes the optimization procedure identical to the one proposed by Li et al. (2020). For our approach, we used the manifold optimization procedure to find a good overlap matrix $S$ for the first 20 epochs, and then completed the training with the generalized Cayley optimization methods for the remaining 180 epochs. The results show that our method improves the prediction accuracy for most combinations of datasets and models, achieving at least the baseline performance. The largest improvements are given by the WRN with Cayley SGD for all four datasets.

Table 2 shows the computation time for each model and dataset for the Cayley approach and the manifold optimization procedure. It also shows the total training time for the baseline Cayley transformation approach and the total training time for our approach. To train the networks in a reasonable time, a GPU is required. Using the NVIDIA GeForce RTX 3080, the duration of a single experiment ranges from 5.42 hours for the VGG model on the CIFAR10 dataset to 30.12 hours for the Wide ResNet on the Tiny ImageNet32. The manifold optimization takes slightly more than twice as long per epoch. This is due to the costly inversion of the $R$ matrix, which is part of the computational graph during backpropagation.

However, our approach remains competitive. Figure 2 shows the test accuracy over the training time. The blue line represents the baseline approach and the red line represents our approach. Note that our approach uses the slower manifold optimization for the first 20 epochs. The change in method is particularly noticeable for the Tiny ImageNet dataset, as it results in a drop in accuracy. However, our test accuracy surpasses the baseline at the first drop in learning rate at 60 epochs, which is before the 100th epoch of the baseline. This shows that finding an appropriate overlap matrix can increase the overall accuracy before half of the training is complete, even with the costly manifold search at the beginning.

Overall, our method shows faster convergence on the training data, competitive test errors that decrease for most combinations of datasets, models, and methods, and it still provides practical advantages due to its behavior over time.

## 6 LIMITATIONS

While the extension of the Caley algorithm to a generalized form is relatively straightforward, it also leads to the problem of defining a generalized form. When we started this work, we used standard hyperparameter optimization strategies such as Bayesian optimization to find $S$. Due to the high dimensionality and structure of the search space, it is difficult to search. Using gradient-based methods, we were not only able to solve this problem within the same optimization regime as training, but we also made it possible to switch between different overlap matrices $S$ during training. Despite empirical success on several datasets, any heuristic optimization strategy is prone to local optima. Furthermore, our experiments show that for CNNs, optimizing $S$ early (as in classical hyperparameter optimization) instead of optimizing both together over the entire training process

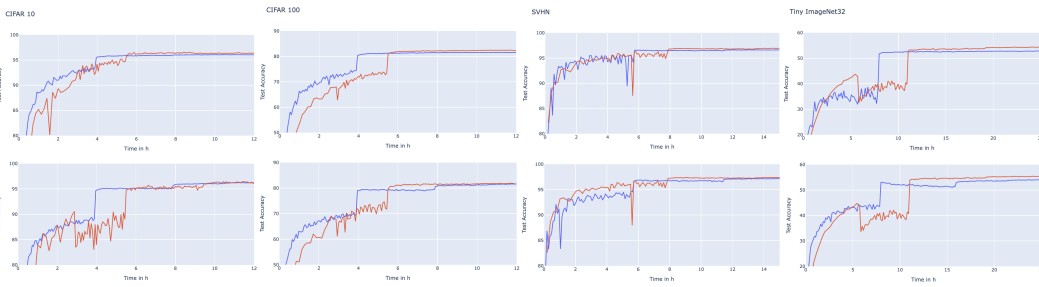

Figure 2: The test accuracy over time for all four datasets with the Wide ResNet. The blue lines represent the baseline approach and the red lines our approach with the manifold optimization and the generalized Stiefel manifold restriction. For each dataset in the upper plots the generalized Cayley SGD was used and for the lower plots the generalized Cayley ADAM.

Table 2: Training time for different methods, models and datasets

|  | | CIFAR | | SVHN | | TINY IMAGENET | |
| --- | --- | --- | --- | --- | --- | --- | --- |
|  | | SGD | ADAM | SGD | ADAM | SGD | ADAM |
| WRN | TIME PER EPOCH (CAYLEY) IN S | 237 | 240 | 344 | 350 | 479 | 488 |
|  | TOTAL TRAINING TIME (200 EPOCHS) IN H | 13,17 | 13,33 | 19,12 | 19,45 | 26,62 | 27,12 |
|  | TIME PER EPOCH (MANI. OPT.) IN S | 516 | 516 | 735 | 735 | 1030 | 1030 |
|  | TIME FOR GEN ST (OURS) IN H | 14.72 | 14.78 | 20.45 | 20.65 | 29.67 | 30.12 |
| VGG | TIME PER EPOCH (CAYLEY) IN S | 87 | 88 | 129 | 131 | 161 | 163 |
|  | TOTAL TRAINING TIME (200 EPOCHS) IN H | 4.83 | 4.89 | 7.17 | 7.28 | 8.94 | 9.05 |
|  | TIME PER EPOCH (MANI. OPT.) IN S | 193 | 193 | 286 | 186 | 346 | 346 |
|  | TIME FOR GEN ST (OURS) IN H | 5.42 | 5.47 | 8.04 | 8.14 | 9.97 | 10.07 |

still gives the best overall convergence. However, optimal training strategies may differ for other datasets, and we have only focused on image data at this stage.

Since we have focused on understanding and extending the previous work of Li et al. (2020), we have also only applied our generalization approach to CNNs. Although we believe that this approach could be applied to other types of neural networks, the kernel structure is unique to CNNs. Furthermore, the success of Stiefel restrictions for other types of neural networks, such as RNNs, is based on their unit norm preservation, which is not given by the generalized approach we used in this paper.

## 7 SUMMARY

Building on the work of Li et al. Li et al. (2020), we propose to generalize the orthogonality constraints of the Stiefel manifold to the generalized Stiefel manifold for training CNNs. This greatly increases the flexibility of the constraint while maintaining positive effects on training. By not only generalizing both Cayley SGD and Cayley ADAM, but also formulating the optimization of the overlap matrix $S$ as a gradient based method, we can improve the training procedure on the CIFAR10, CIFAR100, SVHN, and Tiny ImageNet32 datasets, both in terms of convergence rate and accuracy, by relaxing the constraints imposed by the orthonormal Stiefel manifold.

Our results show that not only orthonormality can help the learning process in neural networks, but also that we can learn other restrictions within the Stiefel manifold that fit the data.

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

# A  PROOF THAT THE CAYLEY TRANSFORMATION STAYS ON THE GENERALIZED STIEFEL MANIFOLD

To prove that $Y(\alpha) \in \text{St}_S(n,p)$ we need to show $Y(\alpha)^T SY(\alpha) = \mathbb{I}$. To do so, we will show that $Q^T SQ = S$, where $Q = (\mathbb{I} - WS)^{-1}(\mathbb{I} + WS)$ with $W$ as an abitrary skew-symmetric matrix. Note that $S$ is invertable and symmetric.

$$
\begin{aligned}
Q^T SQ &= (\mathbb{I} - SW)(\mathbb{I} + SW)^{-1} S (\mathbb{I} - WS)^{-1} (\mathbb{I} + WS) \\
&= (\mathbb{I} - SW) \left[ (\mathbb{I} - WS) S^{-1} (\mathbb{I} + SW) \right]^{-1} (\mathbb{I} + WS) \\
&= (\mathbb{I} - SW) \left[ S^{-1} - W + W - WSW \right]^{-1} (\mathbb{I} + WS) \\
&= (\mathbb{I} - SW) \left[ (\mathbb{I} + WS) S^{-1} (\mathbb{I} - SW) \right]^{-1} (\mathbb{I} + WS) \\
&= (\mathbb{I} - SW)(\mathbb{I} - SW)^{-1} S (\mathbb{I} + WS)^{-1} (\mathbb{I} + WS) \\
&= S
\end{aligned}
$$

The claim $Y(\alpha) \in \text{St}_S(n,p)$ for all $\alpha$ follows directly from $X^T Q^T SQX = X^T SX$

