# OpenReview forum: "Improving CNN training by Riemannian optimization on the generalized Stiefel manifold combined with a gradient-based manifold search"
_ICLR.cc/2025/Conference — Submitted to ICLR 2025_

### Official Review · Reviewer_M6ZN · 2024-10-26

**Soundness:** 3
**Presentation:** 2
**Contribution:** 2
**Rating:** 3
**Confidence:** 5

**Summary:**

This paper proposes a novel approach for training convolutional neural networks (CNNs) using Riemannian optimization on the generalized Stiefel manifold, which introduces a symmetric overlap matrix \( S \) as a hyperparameter. The authors argue that generalizing from the standard Stiefel manifold constraint \( X^T X = I \) to \( X^T S X = I \) allows more flexibility and a larger solution space. The approach employs gradient-based optimization for \( S \) in combination with Riemannian optimization for the CNN parameters and evaluates this method using generalized Cayley SGD and Cayley ADAM optimizers on various datasets. The experimental results show improvements in convergence rates and classification accuracy over traditional Stiefel manifold constraints.

**Strengths:**

1.	The paper introduces the generalized Stiefel manifold as an alternative to traditional orthonormal constraints in CNNs, which has not been widely applied in this context. By treating \( S \) as a tunable hyperparameter, the approach explores a new angle in Riemannian optimization for CNNs.

2.	The experiments present a thorough comparison across datasets, including CIFAR-10, CIFAR-100, SVHN, and Tiny ImageNet32, demonstrating that the proposed manifold constraint leads to faster convergence rates and improved test accuracy in most cases

**Weaknesses:**

1.	While the paper empirically demonstrates the generalized Stiefel manifold's benefits, it lacks a theoretical explanation or proof of why the generalized constraint \( X^T S X = I \) should offer significant advantages over traditional orthonormal constraints in CNN applications. The claim that the generalized manifold “leads to more possible solutions” is not supported by rigorous theoretical arguments. Including a deeper exploration of how this generalization impacts learning dynamics or theoretical properties (e.g., stability or expressivity) would strengthen the work.

2.	The evaluation primarily compares the generalized Stiefel manifold against baseline methods in the paper without broader comparisons with other established regularization techniques such as weight normalization, spectral normalization, or orthogonality regularizations (e.g., Bansal et al., 2018). Such comparisons would provide a more comprehensive assessment of the proposed method's relative performance in CNN optimization.

3.	While the paper uses Bayesian optimization to determine \( S \), it lacks visualization or interpretative analysis showing how different configurations of \( S \) affect convergence behavior. The overlap matrix \( S \) is central to the proposed approach, and further insights into its learned values across datasets or their influence on the optimization path could provide interpretative depth and help readers assess the flexibility and robustness of this method.

4.	The additional complexity introduced by the generalized Stiefel manifold’s optimization process, particularly the inversion of \( S \), doubles the time for training epochs compared to Riemannian optimization without \( S \). While faster convergence is reported, the paper does not address the trade-offs adequately. This additional cost should be discussed in terms of computational efficiency, especially for large-scale datasets.

**Questions:**

How robust is the chosen Bayesian optimization approach for tuning \( S \) across different datasets and models? Would other optimization strategies yield better or more interpretable results?

---

### Official Review · Reviewer_9C19 · 2024-10-31

**Soundness:** 2
**Presentation:** 2
**Contribution:** 1
**Rating:** 3
**Confidence:** 5

**Summary:**

This paper studies training deep neural networks with the generalized Stiefel constraint, and conduct empirical study on small datasets including CIFAR10, CIFAR100, SVHN, and Tiny ImageNet32.

**Strengths:**

Training DNNs with orthonormality constraints was explored in the literature before, and this paper extends such idea to generalized Stiefel constraint and conducted empirical study.

**Weaknesses:**

There are several major concerns.

(1) The technical novelty is rather minimal. This paper considers the generalized Stiefel constraint, $X^{\top} S X = I$, which is an incremental change compared to the regular orthonormality constraint $X^{\top} S X = I$ on the usual Stiefel manifold. Furthermore, there are no theoretical guarantee or analysis on the claims about overall faster convergence rates and/or improved test performance.

(2) The empirical study is only performed on outdated neural network architectures, such as WRN and VGG, and small datasets (CIFAR10, CIFAR100, SVHN, and Tiny ImageNet32). Much more extensive results on modern neural networks, such as vision transformers (ViT, Swin, etc.), on larger benchmarks (at least ImageNet-10k) are expected to justify the empirical claims of this paper.

**Questions:**

See weaknesses.

---

### Official Review · Reviewer_mi1C · 2024-11-02

**Soundness:** 2
**Presentation:** 2
**Contribution:** 2
**Rating:** 3
**Confidence:** 4

**Summary:**

The paper discusses the growing interest in incorporating orthonormality constraints in deep learning, particularly through the use of Riemannian optimization techniques on the Stiefel manifold, which ensures orthonormal parameter matrices in CNNs. While previous studies have shown that orthogonality regularization can improve accuracy and convergence rates, strict enforcement of orthonormality can limit the solution space and hinder performance. To address this, the authors propose a novel approach that generalizes the Stiefel manifold by introducing a flexible overlap matrix, thereby expanding the solution space during training. Their method dynamically optimizes this overlap matrix using gradient-based techniques, promising improved convergence rates and overall accuracy without excessive restrictions on optimization. The introduction situates this work within existing literature and identifies a gap regarding the implementation of generalized Stiefel manifold optimization in deep learning.

**Strengths:**

1. It seems the approach addresses the limitations of strict orthonormality constraints, which have been shown to be disadvantageous in some scenarios.

2. The paper builds upon established Riemannian optimization techniques, presenting a well-structured way for optimizing the overlap matrix.

3.  By addressing the challenges associated with parameter matrix optimization in CNNs, this work has the potential in improving empirical convergence rates and higher test accuracies with advancements in deep learning applications.

**Weaknesses:**

1. While the introduction provides a solid overview, some sections of the paper could benefit from clearer explanations of the mathematical concepts, particularly regarding the generalized Stiefel manifold and the optimization procedures. See questions below

2. The proposed optimization method may introduce additional computational complexity. A more thorough analysis of the computational requirements and efficiency, especially in comparison to existing methods, would be beneficial.

3. Unfortunately large body of the paper is in parallel with the work from paper  Li et al. (2020). The generalized Stiefel manifold and optimization algorithms on it have been well researched.  I see no very significant contribution except for making adaptive generalized Stiefel manifold for network parameters. The paper may not be suitable for ICLR.

4. The implementation details should be released.

**Questions:**

1. It is highly recommended the authors can release their experimental codes e.g. on https://anonymous.4open.science

2. I might miss something. I feel the description between Lines 224 and 231 is just a sketch.  More details should be provided.  When you add R as part of optimization, the original objective function has changed, as RX will appear as the product in the objective, how to control their scale as both are optimization variables.  Based on the current version, it is not clear to me how R_{i+1} was updated.  It is better to provide this formula. Furthermore, instead of presenting algorithms 1 & 2, you may present a modified version with step(s) for updating R_i.

3. Given the time restriction in this urgent call for review, I did not carefully read the experiment section.

---

> ### Comment · Reviewer_mi1C · 2024-11-26
>
> There is no author rebuttals.  It seems the authors withdrew the paper?

---

### Official Review · Reviewer_Awpo · 2024-11-02

**Soundness:** 1
**Presentation:** 1
**Contribution:** 1
**Rating:** 1
**Confidence:** 5

**Summary:**

This paper proposes an approach to improve CNN training by applying Riemannian optimization on a generalized Stiefel manifold, aiming to enhance convergence rates and performance through a dynamic adjustment of the overlap matrix SSS. While the idea of optimizing over a generalized Stiefel manifold could be useful, the paper falls short in several critical areas, both theoretically and empirically.

**Strengths:**

The paper explores a novel approach by generalizing the Stiefel manifold constraint.

**Weaknesses:**

The theoretical novelty is very limited:

1. Riemannian optimization usually retraction and vector transport, instead of an exponential map or parallel transport (L 135). It is not clear why you compare exp and pt with the Cayley map. Until the next section, it comes to me that Cayley is the retraction. until I read the original paper, i realized the Cayley is one of the retractions. All of the above should be clarified and acknowledged in the paper.
2. Eq. (4) is theoretically questionable:
   - Why should S lie in SPD? This will limit the generality.
   - Although $R$ in Eq. (4) lies in $R^{n \times n}$, it is not a Euclidean parameters. How do u respect the non-Euclidean space
   - More importantly, as $R$ changes, the latent space is changing. It is quit weird for the current method to omit this fact. For example, how do you transform the momentum between different manifolds?

3. A very counterintuitive issue is that the authors used momentum but didn’t involve vector transport. through all the algorithms, it's like a straightforward variant of Trivializations [1].

Empirical validation is very unconvincing:
  - The experimental validation is insufficient. The authors only evaluate the method on small datasets (e.g., CIFAR10, CIFAR100, SVHN, Tiny ImageNet32) and use very limited backbones.
  - The comparison method is very limited and far from enough. For instance, as this is a direct variant of trivializations, why the authors miss trivializations is not clear. and Also the comparison with Riemannian Gen-st optimization is missing. I believe they are more, apart from the most natural competitor I mentioned.


[1] Trivializations for Gradient-Based Optimization on Manifolds

**Questions:**

see weakness

---

### Official Review · Reviewer_eVKe · 2024-11-04

**Soundness:** 3
**Presentation:** 2
**Contribution:** 2
**Rating:** 3
**Confidence:** 3

**Summary:**

The paper "Improving CNN training by Riemannian optimization on the generalized Stiefel manifold combined with a gradient-based manifold search" describes an optimization method on the  generalized steifel manifold using a gradient based method

**Strengths:**

Instead of using strict orthonormal constraints (Stiefel manifold), they propose a generalized version with a learnable "overlap matrix S" that:
expands the solution space beyond traditional orthonormal matrices
while maintaining the beneficial properties of orthonormal approaches
and can be optimized using gradient-based methods during training.

**Weaknesses:**

This paper is delta increment of the Li(2020 iclr) paper which proposed the cayley transformation update on the Steifel manifold, this just extends it to the generalized steifel manifold with an overlap parameter S. Once can do a exact extension of the Key steps in Li to this paper by adding extra terms related to the S matrix. This make the theoretrical contribution weak.

The experiments seem a bit forced. Why is an orthogonality regularizer needed at all? Does this make the model work better for other deep network models on these well known datasets. Limited comparsion with other approaches dont support that argument

**Questions:**

na

---

### Meta-Review · Area_Chair_qUT4 · 2024-12-19

**Metareview:**

This paper mainly focuses on improving CNN training by Riemannian optimization on the generalized Stiefel manifold. It achieves faster convergence rates and performance on several commonly adopted datasets. It receives all five negative ratings, including one strong reject and four reject. Reviewers are concerned about the limited novelty, incremental contribution, insufficient analysis, etc. Specifically, based on the method of  Li(ICLR2020), this paper mainly extends it to the generalized Stiefel manifold with an overlap parameter S, where the contribution is somewhat limited and incremental. The concerns still exist since the authors also do not present a response during the rebuttal phrase. I think the current manuscript does not meet the requirements of this top conference. I suggest the authors carefully revise the paper and submit it to another relevant venue.

**Additional Comments On Reviewer Discussion:**

The authors do not provide a response.

---

### Decision · Program_Chairs · 2025-01-22

Reject